# Sustainable Approach to Eradicate the Inhibitory Effect of Free-Cyanide on Simultaneous Nitrification and Aerobic Denitrification during Wastewater Treatment

Ncumisa Mpongwana [1], Seteno K. O. Ntwampe [1,2,*], Elizabeth I. Omodanisi [1,3], Boredi S. Chidi [1] and Lovasoa C. Razanamahandry [1,4,5]

1　Bioresource Engineering Research Group (BioERG), Faculty of Applied Sciences, Cape Peninsula University of Technology, P.O. Box 652, Cape Town 8000, South Africa; mpongwanancumisa@yahoo.com (N.M.); lizzy.omodanisi@gmail.com (E.I.O.); boredi2002@gmail.com (B.S.C.); tantely1989@gmail.com (L.C.R.)
2　Department of Chemical Engineering, Cape Peninsula University of Technology, P.O. Box 652, Cape Town 8000, South Africa
3　Department of Biological Sciences, Mountain Top University, Ogun State 110106, Nigeria
4　UNESCO UNISA Africa Chair in Nanoscience's/Nanotechnology Laboratories (U2AC2N), College of Graduate Studies, University of South Africa (UNISA), Muckleneuk Ridge, P.O. Box 392, Unisa 0003, South Africa
5　Nanosciences African network (NANOAFNET), Materials Research Group (MRG), iThemba LABS—National Research Foundation (NRF), 1 Old Faure Road 7129, P.O. Box 722, Somerset West, Western Cape Province, Cape Town 8000, South Africa
*　Correspondence: NtwampeS@cput.ac.za; Tel.: +27(0)21-460-3172

**Abstract:** Simultaneous nitrification and aerobic denitrification (SNaD) is a preferred method for single stage total nitrogen (TN) removal, which was recently proposed to improve wastewater treatment plant design. However, SNaD processes are prone to inhibition by toxicant loading with free cyanide (FCN) possessing the highest inhibitory effect on such processes, rendering these processes ineffective. Despite the best efforts of regulators to limit toxicant disposal into municipal wastewater sewage systems (MWSSs), FCN still enters MWSSs through various pathways; hence, it has been suggested that FCN resistant or tolerant microorganisms be utilized for processes such as SNaD. To mitigate toxicant loading, organisms in SNaD have been observed to adopt a diauxic growth strategy to sequentially degrade FCN during primary growth and subsequently degrade TN during the secondary growth phase. However, FCN degrading microorganisms are not widely used for SNaD in MWSSs due to inadequate application of suitable microorganisms (*Chromobacterium violaceum*, *Pseudomonas aeruginosa*, *Thiobacillus denitrificans*, *Rhodospirillum palustris*, *Klebsiella pneumoniae*, and *Alcaligenes faecalis*) commonly used in single-stage SNaD. This review expatiates the biological remedial strategy to limit the inhibition of SNaD by FCN through the use of FCN degrading or resistant microorganisms. The use of FCN degrading or resistant microorganisms for SNaD is a cost-effective method compared to the use of other methods of FCN removal prior to TN removal, as they involve multi-stage systems (as currently observed in MWSSs). The use of FCN degrading microorganisms, particularly when used as a consortium, presents a promising and sustainable resolution to mitigate inhibitory effects of FCN in SNaD.

**Keywords:** denitrification; free cyanide; nitrification; simultaneous nitrification and aerobic denitrification; wastewater treatment

## 1. Introduction

Excessive nitrogenous compounds in wastewater discharged into water bodies such as rivers can result in dissolved oxygen (DO) depletion and eutrophication in the receiving rivers [1]. Due to governmental regulations in place to regulate treated wastewater discharge standards, it is important that wastewater containing a high concentration of nitrogenous compounds must be treated prior to discharge [2]. This type of wastewater can be treated by biological processes such as simultaneous nitrification and aerobic denitrification (SNaD) or physico-chemical processes such as ammonium stripping, chemical precipitation of ammonia, electrochemical conversion, and many other treatment technologies [3].

However, biological treatment of total nitrogen (TN) laden wastewater via traditional methods, i.e., nitrification and subsequent anoxic denitrification in a two-step set-up, is the desired method for treatment of TN in generic municipal wastewater sewage systems (MWSSs) because these methods are efficient at a larger scale. Overall, biological treatment uses the metabolic activity of living organisms in consortia for pollutant removal, with microorganisms such as bacteria primarily being used in an agglomerated symbiotic biological potpourri of reactions in sequential or parallel processes. Nonetheless, biological treatment methods are not always suitable to treat some industrial wastewater due to the toxicity of organic and other substances therein [4], which reduces these methods' efficiency.

An example is coking wastewater, which contains a high concentration of free cyanide (FCN), which decomposes to ammonium-nitrogen, nitrates, and nitrites, herein referred to as TN and phenolics. Such wastewater, if treated in an inefficient primary process, would culminate in the inhibition of biologics of downstream processes such as nitrification and denitrification, resulting in the disposal of partially treated wastewater still containing a high concentration of TN. Moreover, when primary and secondary wastewater treatment processes experience increased toxicant loading such as FCN from industrial processes in combination with secondary pollutants, e.g., phenolics or heavy metals, the discharged FCN containing wastewater would further contribute to receiving surface water pollution, a challenge which is further exacerbated by runoff from agricultural operations whereby the use of cyanogen-based pesticides is still in practice, especially in developing countries. In certain instances, the remedial strategy implementable to minimize FCN inhibition toward primary and secondary processes such as nitrification and denitrification sometimes involves the use of adsorbents such as activated carbon as a sorbent [5] for FCN adsorption. Conversely, the application of physical processes such as activated carbon adsorption, would incur additional operational costs associated with the procurement of the adsorbent and its disposal, including regeneration if it is to be used in multi-cycle operations.

Additionally, the use of sorbents such as activated carbon is less effective in eliminating the inhibitory effect of FCN in nitrification and subsequent denitrification, particularly when periodic spillovers to these processes downstream occur and when inadvertent adsorption–desorption processes in the primary process occur due to process conditions variation, including wastewater quality changes. This can also be due to the low absorption capability of poor quality activated carbon used in some operations and because the affinity of FCN to activated carbon is low [6]. Therefore, it is prudent to invest in and investigate a sustainable method to eliminate the inhibition of FCN towards nitrification and denitrification. Kim et al. [7] suggested the use of FCN degrading bacteria to eliminate cyanide inhibition towards nitrification and subsequent denitrification. Furthermore, although both nitrification and anoxic denitrification occur as separate processes at an industrial scale [8], several research studies have indicated the use of simultaneous nitrification and aerobic denitrification (SNaD), which effectively culminates in the integration of a traditional two-staged process into a single-stage process [9,10] with an added benefit of having a reduced footprint; albeit, there is minimal literature on the utilization of SNaD as a sustainable process in which FCN degrading bacterial consortia are used, a practice yet to be adopted at an industrial scale.

## 2. Multi-Stage Nitrification and Subsequent Denitrification: An Obsolete Technology

The secondary treatment in wastewater uses biological processes due to their cost-effectiveness and environmental benignity compared to physical treatment technologies, which are expensive and produce toxic by-products including waste material. Biological treatment plays a crucial role during nutrient removal and for the prevention of eutrophication in receiving water bodies [11]. Nitrification and subsequent denitrification are among the important biological processes that are currently being successfully employed in MWSSs for the removal of TN [4]. Generally, the process of TN removal is initiated with aerobic ammonium-nitrogen ($NH_4$-N) oxidation in a two-step process with the first step being nitritation and the second being nitratation. During nitritation, ammonia-oxidizing bacteria (AOB) oxidize $NH_4$-N to $NH_2OH$ through ammonia monooxygenase (AMO) biocatalysis; the $NH_2OH$ is oxidized further into $NO_2^-$ through hydroxylamine oxidoreductase (HAO) [11].

This process is known as nitrification through the nitrite route and is ideal as it reduces carbon source requirements by up to 40%, thus reducing costs associated with carbon source utilization. The second step involves the oxidation of $NO_2^-$ into $NO_3^-$ by nitrite-oxidizing bacteria (NOB) catalyzed by nitrite reductase (NIR) [12,13]. Although nitrification is successfully applied in MWSSs for TN removal, it is a highly sensitive process [14]. The effluent from nitrification is further processed in an anaerobic reactor for anoxic denitrification, whereby microorganisms oxidize nitrates into gaseous nitric oxide (NO) and nitrous oxide ($N_2O$) bio-catalytically facilitated by nitric and nitrous oxide reductases (NorB/NosZ). Furthermore, these exhaust gasses are reduced into di-nitrogen ($N_2$) gas, which acts as a terminal acceptor for electron transport phosphorylation under anaerobic conditions [15]. Anoxic denitrification also catalyzes the formation of the N–N bond from process (denitrification) intermediates, i.e., NO and $N_2O$ [16]. Nitrification and denitrification pathways as well as the enzymes involved can be depicted summarily by Figure 1.

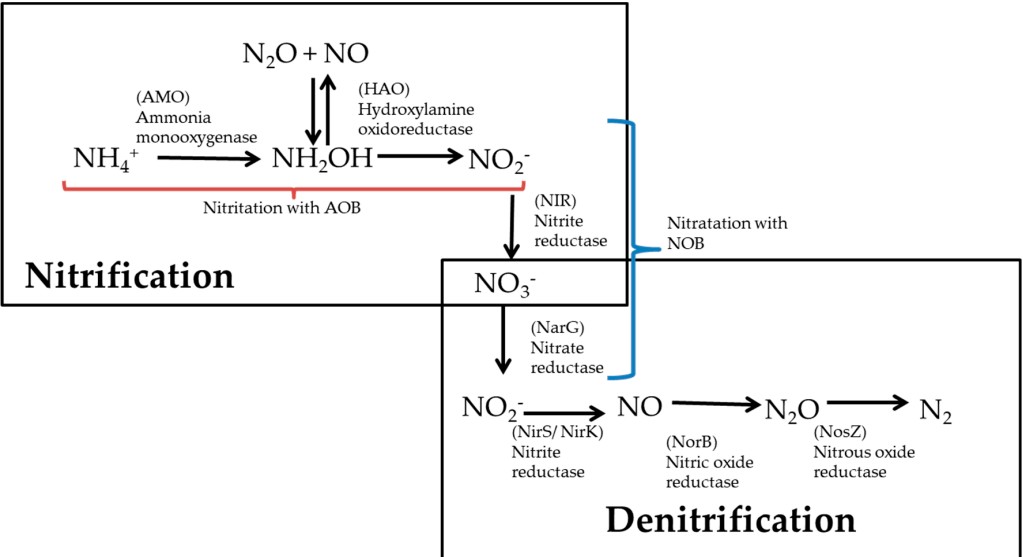

**Figure 1.** Diagram representing nitrification and subsequent denitrification.

Denitrification was also proven to occur under aerobic conditions [17]; hence, this development offered a possibility for SNaD that is more cost-effective for TN removal than the traditional nitrification and the subsequent denitrification processes currently used in MWSSs [9,10]. Some of the microorganisms that were proven to carry-out denitrification under completely aerobic conditions include: *Pseudomonas alcaligenes* AS-1, *Pseudomonas* species (sp.) 3–7, *Pseudomonas* sp. *Rhodoferax ferrireducens*, *Agrobacterium* sp. LAD9, *Rhodococcus* sp. CPZ 24, *Bacillus subtilis* A1, *Pseudomonas stutzeri* YZN-001, *Acinetobacter calcoaceticus* HNR, *Bacillus methylotrophicus* L7, *Diaphorobacter* sp., *Acinetobacter* sp. Y1, *Acinetobacter junii* YB, and *Marinobacter* sp [9,10,18].

In addition, a number of other aerobic denitrifying bacteria have been isolated and identified, e.g., *Paracoccus* (Micrococcus) *denitrificans*, *Hyphomicrobium* strains, *Hyphomicrobium vulgare*, *Moraxella* species, and *Kingella denitrificans* [19]. Although nitrification and denitrification were proven to be sustainable methods for treating TN, more research was done to improve these processes such that they are more sustainable, more cost-effective, and easy to operate. Some of the important genes and the processes that are responsible for nitrification and denitrification are highlighted in Table 1.

**Table 1.** Genes responsible for nitrification and denitrification and their functions [16].

| Category of Affected Process | Gene or Locus | Encoded Gene Product and Their Functions |
|---|---|---|
| Regulation | *anr* | Fumarate and nitrate reductase (FNR)-like global redox regulator for the expression of denitrification genes. |
| | *Dnr, fnrD* | FNR-like regulator that affects the expression of *nirS* and *norCB*. |
| | *Fixk₂* | FNR-like regulator that affects anaerobic growth on nitrate. |
| | *fnrP* | FNR-like regulator that affects the expression of *narGH*. |
| | *narL* | Nitrate responsive transcription factor of *Pseudomonas* of a *narXL* two- component system. |
| | *nirI* | A membrane protein with similarity to *NosR* affects *nirS* expression. |
| | *nirR* | *Pseudomonas* locus that affects the synthesis of *nirS* and *LysR* regulator. |
| | *nirY (orf 286)* | FNR-like regulator that affects expression *nirS* and *norCB* in *Paracoccus* and *Rhodobacter sp*. |
| | *nnrS* | Activate transcription of nirK and nor genes in *Rhodobacter sphaeroides*. |
| | *nosR* | Membrane-bound regulator required for transcription of *nosZ*. |
| | *rpoN* | Sigma factors affect denitrification in *Ralstonia eutropha* |
| Nitrate respiration | *narD* | Plasmid bone locus for eutropha respiratory nitrate reduction. |
| | *narG* | α-subunit of nitrate reductase respiration that binds to molybdopterin guanine dinucleotide (MGD). |
| | *narH* | B-subunit of nitrate reductse respiration that binds to Fe-S cluster. |
| | *narI* | Cytochrome b subunit of respiratory nitrate reductase. |
| | *narJ* | Protein required for nitrate reductase assembles. |
| Periplasmic nitrate reduction | *napA* | The large subunit of periplasmic of nitrate reductase that binds to bis- molybdopterin guanine dinucleotide (MGD) and Fe-S cluster. |
| | *napB* | Small subunit of periplasmic of nitrate reductase, a diheme cytochrome c. |
| | *napD* | Cytoplasmic protein with presumed maturation function, homologous to *Escherichia Coli napD* (YojF). |
| | *napE* | Putative monotopic membrane protein; there are no known homologs. |
| Nitrite respiration | *nirB* | Cytochrome c$_{552.}$ |
| | *nirC* | Monoheme cytochrome c with a putative function in *NirS* maturation. |
| | *nirK, nirU* | Cu-containing nitrite reductase. |
| | *nirN orf507* | It affects anaerobic growth and in-vivo nitrite reduction, similar to *NirS*. |
| | *nirQ* | Gene product that affects catalytic functions of *NirS* and *NorCB*. |
| | *nirS (denA)* | Cytochrome cd, nitrate reductase. |
| Heme D$_1$ Biosynthesis | *nirD* | Gene product affects heme D. Biosynthesis or processing. |
| | *nirE* | S-Adenosyl-l-Methionine uropophyrinogen III methyltransferase. |
| | *nirF* | Needed for heme D biosynthesis and processing; similar to *NirS*. |
| | *nirG* | Gene product affects heme D. Biosynthesis or processing. |
| | *nirH* | Gene product affects heme D. Biosynthesis or processing. |
| | *nirJ, orf393* | Needed for heme D biosynthesis and processing; similar to PqqE, *NifB*, and *MoaA*. |
| | *nirL* | Gene product affects heme D. Biosynthesis or processing. |
| NO respiration | *norB* | Cytochrome b subunit of NO reductase. |
| | *norC* | Cytochrome c subunit of NO reductase. |
| | *norD, orf6* | Affect availability under denitrifying conditions. |
| | *norE, orf2, orf175* | Membrane protein: homologous with COX III. |
| | *norF* | Affect NO and nitrite reductase. |
| | *norQ* | Affect *NirS* and *NorCB* function; homolog of *NirQ*. |
| N$_2$O respiration | *Fhp* | *R. eutropha* flavohemoglobin affects N$_2$O and NO reduction. |
| | *nosA, oprC* | Channel-forming outer membrane protein; Cu-processing for *NosZ*. |
| | *nosD* | Periplasmic plastic involved in Cu insertion into *NosZ*. |
| | *nosF* | ATP or GDP binding protein involved in Cu insertion into *NosZ*. |
| | *nosL* | Part of nos gene cluster; putative outer membrane lipoprotein. |
| | *nosX* | Affect nitrous oxide reduction in *Sinorhizobium meliloti*. |
| | *nosY* | Inner membrane protein involved in Cu processing for *NosZ*. |
| | *nosZ* | Nitrous oxide reductase. |
| Electron transfer | *azu* | Azurin. |
| | *cycA* | Cytochrome C$_2$ (C$_{550).}$ |
| | *napC* | Tetraheme cytochrome c; homologous to *NirT*. |
| | *nirM (denB)* | Cytochrome C$_{551.}$ |
| | *nirT* | Putative membrane-anchored tetraheme c-type cytochrome. |
| | *paz* | Pseudoazurin. |
| Functionally unassigned | *Orf396* | A putative 12 span membrane protein of *Pseudomonas stutzeri* homologous to NnrS. |
| | *nirX* | A *Paracoccus* putative cytoplasmic protein; homologous to *NosX*. |
| | *orf7, orf63* | *Pseudomonas* gene downstream of dnr and *fnrD*. |
| | *orf247* | Putative member of the short-chain alcohol dehydrogenase family. |

## 3. Recent Advances in Nitrification and Denitrification Processes: Future Perspectives

Denitrification was believed to occur under completely anoxic conditions [9,10], while nitrification emerged as an aerobic process [18]. Furthermore, the growth of nitrifiers depends on DO, which is lethal to traditional denitrifiers. Conversely, some microorganisms that are capable of heterotrophic nitrification and aerobic denitrification have been reported; hence, SNaD has recently drawn attention due to its potential to reduce cost related to the second anoxic tank whereby denitrification would have occurred [17,20–24].

Additionally, aerobic denitrification can also regulate and maintain the pH in the reactor since nitrification causes acidification [18]. Aerobic denitrification occurs in two ways—the first mechanism is due to aerobic respiration aided by an enzyme known as periplasmic nitrite reductase (NAR)—see Figure 2A. This enzyme is essential for the conversion of nitrate to nitrite under aerobic conditions [17]. However, due to the sensitivity of $N_2O$ reduction enzymes to DO, a significant amount of NO and $N_2O$ are emitted to the environment [25]. The second mechanism is through the transfer of DO into the activated sludge flocs for nitrification, which results in the diffusion competition whereby the DO consumption becomes greater in the outer zone of the floc, thus reducing DO penetration into the interior of the floc and leading to an anoxic zone in the center of flocs (see Figure 2B), which is suitable for denitrification [26].

The increase in operational costs resulting from the dosing of synthetic and industrial-grade chemicals in the biological MWSSs [27] was a major driver for SNaD development in a signle stage, low-cost and environmentally benign process. This can involve the use of agricultural waste to sustain microbial growth during SNaD. The ability of SNaD microorganisms to grow onto agricultural waste is due to the availability of trace elements, micro and macro-nutrients on the waste itself, which can serve as readily available nutrient sources and a biomass immobilization matrix for microbial proliferation [28,29].

Mekuto et al. [30] also proved that agricultural waste can be used as a sole supplementation source of microbial growth during biodegradation of FCN-TN. However, the microorganism or consortia may also convert some unintentional sources within the agricultural waste into undesirable and desirable biomolecules such as citric, lactic, succinic acid, and alcohols [31] during wastewater treatment. Furthermore, these biomolecules can also cause fluctuations in the wastewater pH, which will eventually lead to the inhibition of some essential microbial populations that are responsible for the biological processes in the MWSSs.

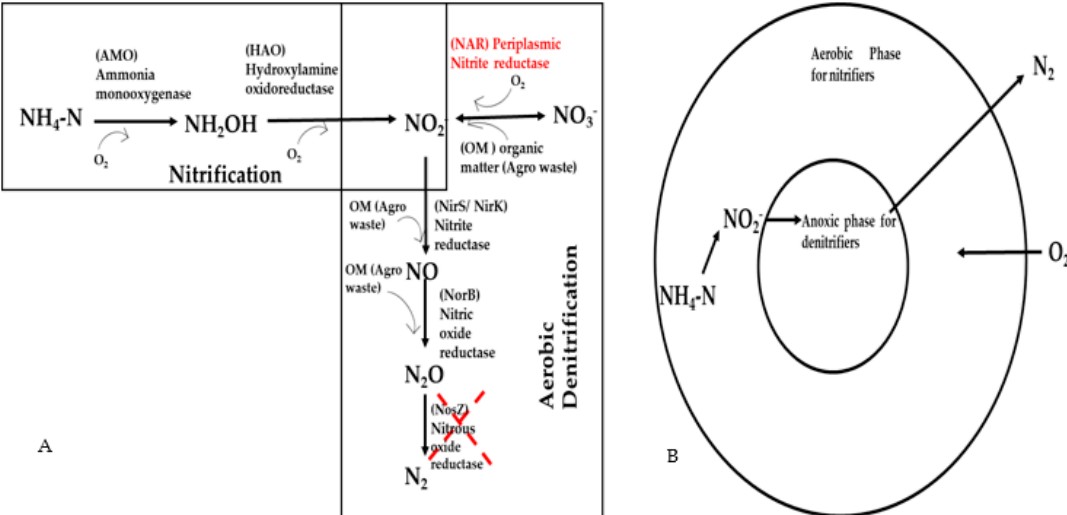

**Figure 2.** Diagram representing (**A**) different simultaneous nitrification and aerobic denitrification mechanisms as well as simultaneous nitrification and aerobic denitrification via nitrite route. (**B**) Representation of floc in activated sludge with aerobic and anoxic zone.

Additionally, activated sludge processes are known to be relatively high energy-consuming processes that lead to the escalation of plant operational costs, thus making biological processes less sustainable, especially in developing countries. This has led to strategies aiming at improving the operational conditions of these biological processes [32] by altering reactor configurations. Consequently, it has been reported that 2% of all electrical power in the USA is used by MWSSs, and a further 40–60% of all the energy is used for aeration and mechanical devices such as stirrers and diffusers (including nozzles), with only 5–25% of supplied air embedded oxygen being successfully transferred to the wastewater as DO and the rest becoming only pneumatically expunged oxygen in bubbles purged without transfer [33].

As a result, the replacement of conventional activated sludge systems by cost-effective reactors was eminent for lowering operational costs and thus the adherence of the sequence batch reactor (SBR). Initially, the SBR was shown to be cost-effective for nitrification and sequential anoxic denitrification; hence, such reactors are easily adaptable to operate in a different mode and allow for both nitrification and aerobic denitrification to occur in the same tank, resulting in SNaD [34,35]. Moreover, SBR is popular and is of interest since it was proven to save up to 60% of the expenses required by conventional activated sludge processes whilst being highly versatile and efficient. Additionally, it has a short retention time compared to other conventional activated sludge processes, which require 3–8 h of continuous aeration [32]. In addition, other reactors, including the membrane biofilm reactors (MBfRs), are also known to be cost-effective and highly efficient [33,36]. Process interchangeability can provide for process improvement. A summary of process configurations interchangeability for nitrification and denitrification is denoted in Figure 3; whereby a demonstration of how a sequence batch reactor (SBR) can be interchangeably replaced with a membrane biofilm reactors (MBfR) for high process efficiency

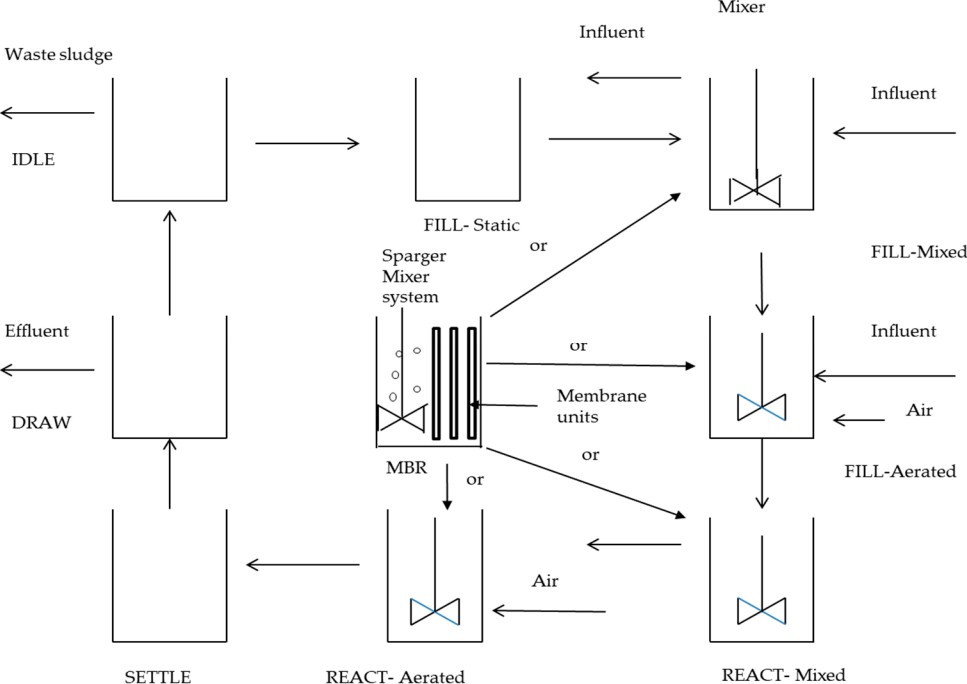

**Figure 3.** Principles of sequence batch reactor (SBR) and how interchangeable they can be with membrane biofilm reactors (MBfR) systems.

Modelling is another important aspect of sustaining a smooth operation of a process, e.g., wastewater treatment. Most MWSS plants are process controlled using advanced process control models and systems. For SNaD in SBR type processes, modelling has been applied to predict and control environmental process conditions and wastewater quality for the SNaD to succeed. Different mathematical models have been used to predict oxidation of TN; however, these models fail to explore

metabolic activities and networks of the microbial populations used during SNaD [37–39]. A recently proposed mathematical exposition to explain SNaD is illustrated in Kanyenda et al. [40].

These mathematical models also fail to accurately address the metabolic networking of microbial populations responsible for SNaD [41]. Thus, they cannot be used to describe biological processes used in MWSSs, since biological processes rely on metabolic networking of microbial populations, particularly for consortia-catalyzed systems.

*Overall Remarks on Simultaneous Nitrification and Aerobic Denitrification (SNaD): Advances and Limitations*

All these improvements have contributed to a significant difference in the smooth operation of nitrification and denitrification for TN removal. Moreover, these improvements have also made a considerable reduction in the operational cost of these processes. Nevertheless, with all the efforts made to advance nitrification and denitrification, MWSSs still face challenges—they are easily inhibited by many contaminants present in the wastewater, resulting in a negative impact on their operation and rendering the overall processes ineffective. Hence, efforts have been made to address such challenges.

## 4. Challenges in Simultaneous Nitrification and Aerobic Denitrification (SNaD) Processes

The major challenges SNaD systems are currently facing are the slow growth rate and the sensitivity to temperature, pH, DO concentration, and toxicants, which negatively affect nitrifying and denitrifying organisms [42,43]. Additionally, high shear stress resulting from aeration can also result in the slow growth of nitrifying and denitrifying microorganisms [44], causing excessive biomass wash-out during wastewater treatment and resulting in reduced TN removal efficiency and SNaD failure [45].

This could ensure SNaD susceptibility to inhibition by toxicants and heavy metals present in the wastewater. High concentrations of heavy metal are usually found in nitrogen-rich wastewaters from anaerobic digestates, e.g., anaerobically digested piggery and dairy slurries [46]. Although heavy metals affect SNaD, they are required in small quantities to enhance microbial growth and stimulate the activity of microorganisms by stimulating enzymes and co-enzymes that play an important role in SNaD, e.g., copper and molybdenum, which are constituents of nitrite reductase and nitrite oxidoreductase, respectively, while other known enzymes involved in SNaD depend on other heavy metals such as nickel-dependent hydrogenase, ATP-dependent zinc metalloprotease FtsH 1, and zinc-containing dehydrogenase [46].

Although minute amounts of heavy metals such as Fe, CU, Co, Ni, and Zn are essential in wastewater treatment, their toxicity towards nitrifying and denitrifying microorganisms is mainly influenced by metal speciation, sludge health sloughing, and the type of reactor used [47]. Moreover, denitrification inhibition by high concentrations of nitrate in wastewater also affects the metabolism of nitrifying and denitrifying organisms. Another challenge that hinders the practicality of SNaD is the inhibition of denitrifiers by DO. Additionally, operational, maintenance, and process control strategies can produce better reactor performance in general wastewater systems but the same strategies can also hamper SNaD, especially under rudimentary process control conditions that facilitate undesirable influent loadings and environmental conditions [48].

Another challenge with SNaD is the elongated start-up and stabilization period, with the $NH_4$-N and $NO_2^-$ concentrations within the system able to affect the growth of SNaD organisms by stunting the microbial community proliferation during this period. Low $NH_4$-N and $NO_2^-$ concentrations can also result in substrate limitation and can thus lead to a low growth rate of the SNaD microbial populations. Two start-up procedures for SNaD are known to exist, with the first involving directed evolution of the SNaD microorganisms by adaption to increasing $NH_4$-N and $NO_2^-$ concentrations. The second procedure involves the physical inoculation with anoxic denitrifying consortium after the primary (nitrification) step of the SNaD has been initiated. Then, the nitrification and the partial aerobic denitrification in SNaD can thereafter ensue such that they are well established in a one process unit [49]. The inhibition of SNaD by FCN is another common challenge, as FCN has been reported

to possess the highest inhibitory effect toward SNaD; furthermore, some microorganisms suited for SNaD have been reported to use FCN as a nitrogenous source [50].

## 4.1. Prevention of Biomass Washout During the Start-Up of SNaD

Environmental engineers have been making efforts to reduce the start-up time of SNaD microorganisms in order to reduce biomass washout and maintain the TN removal efficiency [44]. Different reactors with low retention times have been designed and studied, including the fluidized bed reactor, the membrane reactor, the gas lift reactor, the rotating biological reactor, and the up-flow anaerobic sludge blanket; however, a portion of biomass is still washed out with the effluent in all these systems, particularly for unstable periods, due to the cases overloading to increase wastewater treatment through-put rates, which induces biomass sloughing and flotation and which results in wash-out [51].

The sequencing batch reactor has been found to be the more suitable reactor for the growth of SNaD microorganisms and is efficient in biomass retention. The possibility of immobilization of SNaD microorganisms as a biofilm on the surface carriers has also been explored as another alternative to reducing biomass washout. The materials that have been well studied as surface carriers include zeolite, polyethylene sponge strips, porous non-woven fabrics, novel acrylic resin materials, bamboo charcoal, and polyurethane spheres [52].

Szabó et al. [45] also showed that by gradually improving biomass health, settling can also reduce SNaD washout. Parameters such as changing DO aeration strategy and contaminant load adaptation during the early stage of the start-up as well as the availability of soluble chemical oxygen demand (COD), which can readily be consumed prior to the commencement of the aeration phase at a low temperature (20 °C) and a neutral pH, can greatly affect the retention of biomass in SNaD processes. These parameters have been studied in order to optimize the functionality of the SNaD [53,54].

Furthermore, washout can be prevented by toxicant removal by the addition of psycho-chemical pre-treatments, which might involve chemical precipitation, adsorption, ion exchange, and electrochemical deposition.

Additionally, these psycho-chemical pre-treatments may result in additional process operational costs; hence, it is imperative to shift to a biotechnologically sustainable approach to avoid slow startup and improve biomass retention by controlling the inhibition of SNaD organisms by toxic pollutants present in the wastewater. FCN degrading bacteria have been reported to have a fast-growing rate; hence, they can provide a practical solution to the inhibition of FCN and eliminate challenges associated with slow growth of SNaD microorganisms [50].

## 4.2. Inhibition Mechanism of Simultaneous Nitrification and Aerobic Denitrification by Pollutants

With all the efforts that have been made to improve SNaD, this process still faces challenges, such as inhibition by toxic pollutants. This is due to the slow growth of NOB, making SNaD prone to inhibition. It has been shown that SNaD is more sensitive to FCN and phenol loading; as little as 1–2 mg/L of hydrogen cyanide (HCN), all of which could result in complete inhibition of metabolic functions of both AOB and NOB, even in consortia bio-catalyzed SNaD. The presence of high concentrations of FCN in the MWSSs can render the secondary treatment processes ineffective subsequent to the disposal of wastewater containing a high concentration of TN, resulting in the deterioration of the MWSS's effluent quality [55,56]. Different inhibition mechanisms of SNaD by different pollutants have been reported. Primary inhibition involves the deactivation of the actions or the activity of ammonia monooxygenase (AMO), which is an important enzyme in the primary step of nitrification, through the inhibition of the respiration system of the microorganism by exogenous ligands that attach to the heme protein (His-Fe2+-His) [57]. The heme protein is required for mediation of the redox processes and respiration, which aid in the reduction of dissolved compounds by bacteria in MWSSs [58].

Secondary inhibition is through the binding of an inhibitor to the active site of the enzyme prohibiting the binding of the substrate (i.e., $NH_4$-N), thus inhibiting its oxidation. Another inhibition

phenomenon involves the removal of the AMO-Cu co-factor through chelation, culminating in the formation of an unreactive complex and rendering the whole SNaD process ineffective. The presence of Cu co-factors has been found to play a crucial role in the activity of AMO, which affects the oxidation of $NH_4$-N. The last enzymatic inhibition involves substrate oxidation, which causes the substrate to be highly reactive, resulting in the premature excretion of the AMO as a secondary metabolite [58]. FCN has been proven to greatly inhibit SNaD in activated sludge systems, primarily due to inadequate AMO activity [5]. FCN inhibits nitrification and denitrification by acting as an exogenous ligand, which binds into His-$Fe^{2+}$-His in three sequential steps, which are: (1) the ionic exchange of the endogenous ligand; (2) the formation of a reactive penta-coordinated species; and (3) the binding of the external ligand [59].

Additionally, Inglezakis et al. [60] showed that the specific $NH_4$-N uptake rate is less inhibited compared to a specific oxygenation rate. It was thus concluded that the autotrophic biomass was less sensitive to FCN than heterotrophic biomass. The inhibition of SNaD by FCN has been widely studied by many, including Kim et al. [55]. Moreover, efforts have been made to try to eradicate SNaD inhibition by using techniques such as the application of pretreatment systems with adsorption processes and the addition of a step whereby microorganisms are used to treat FCN to an acceptable concentrations that has a lessened impact on TN removal subsequent to SNaD.

### 4.3. FCN Wastewater in Municipal Wastewater Sewage Systems (MWSSs) and Its Impact on Nitrification and Denitrification: A Culture of Illegal Wastewater Dumping

FCN is a toxic carbon-nitrogen radical found in various inorganic and organic compounds, some of which are used on an industrial scale. A common form of FCN is hydrogen cyanide (HCN), which can be an odorless gas characterized by a faint, bitter, almond-like odor [61,62]. Cyanide can be found in different forms depending on the pH; at high pH, it is found as an ion of FCN and evaporates as HCN at neutral pH, pKa 9.2. Additionally, FCN has a high affinity for metals and thus can form complexes with metals found in nature even when released in agricultural soil [63]. These metal FCN complexes can be categorized into two categories—weak acid dissociable (WAD) and strong acid dissociable (SAD) FCN complexes [8,64]. Microorganisms and animals also produce minute quantities of FCN as a protection mechanism, e.g., cassava, corn, and lima beans, forages (alfalfa, sorghum, and Sudan grasses), and horticulture plants (ornamental cherry and laurel). FCN is often released as a nitrogenous source when the plant is also under stressed environmental conditions [65].

As such, FCN enters MWSSs via illegal disposal of wastewater, mostly from different industries [50], and as runoff from the disposal of FCN containing agricultural wastes in landfills, the use of FCN containing pesticides, and through the use of FCN containing tar salts. FCN is known for being a metabolic inhibitor of many microorganisms, and as little as 0.3 mg/L can result in the loss of biological activity in microorganisms [66]. It alters the metabolic functions of the organism by forming a stable complex with transient metals that plays a significant role in the functioning of some proteins, including micro and macro-metallo contents within cells, which play an important role in nutritional sustenance of biomass intended for FCN bioremediation [62]. In the wastewater treatment process catalyzed by biomass, this can result in the inhibition of SNaD [8,67].

Some microorganisms produce minute quantities of cyanide for defensive purposes [68]; albeit, they are able to carry-out most metabolic functions in the presence of low FCN. Furthermore, some organisms are able to survive FCN expressing specialized enzymes for the degradation of FCN into $NH_4$-N and $CO_2$ through nicotinamide adenine dinucleotide (NADH)-linked cyanide monooxygenase (CNO), including enzymes such as nitralase and cyanide hydratases (CHTs) [68,69]; these can be rendered ineffective by chelation reaction-side blockages and promotion of redundancies in the overall functionality of the bio-catalysis process.

FCN has been reported to be a highly poisonous compound known to man [70], and it is hyper-toxic under aerobic conditions, which would mean higher toxicity for aerobic organisms used in SNaD. It inactivates the respiration of many microorganisms by binding to the cytochrome-c oxidase [71]. However, some microorganisms have developed a metabolic FCN detoxification

mechanism. These mechanisms have been studied in numerous microorganisms, which culminated in an interest in the research community in SNaD, even under toxicant loading and particularly under FCN loading. Studies have also shown that these microorganisms can either use FCN primarily as a nitrogenous or as a carbon source by converting it to $NH_4$-N and $CO_2$ through NADH-linked cyanide oxygenase [72]. FCN degradation in aerobic conditions can be expressed as highlighted in Equation (1).

$$2HCN + O_2 \overset{Enzyme}{\Rightarrow} 2HCNO \tag{1}$$

whereby the hydrogen cyanate is therefore hydrolyzed into $NH_4$-N and $CO_2$ (Equation (2)):

$$CNO + 2H_2O \rightarrow NH_4^+ + O_2^- \tag{2}$$

## 5. Current Solutions to the Challenges in Simultaneous Nitrification and Aerobic Denitrification (SNaD)

*5.1. Physical Process Used as Remedial Strategy to Decrease the Inhibitory Effect of FCN on SNaD*

Chemical methods have been employed to decrease the concentrations of FCN prior to SNaD. One of the few chemical methods used includes alkaline chlorination oxidation. This method is a preferred chemical method since it is highly effective; however, alkaline chlorination (and thus oxidation) results in undesirable byproducts and produces excess hypochlorite, which is a toxicant. Chemical precipitation by ferrous sulfate is another method that is preferred for FCN removal due to its cost-effectiveness and availability of the salt, but it produces large quantities of toxic sludge. Ion exchange can also be used to lower FCN concentration, although it is difficult to operate and has high input costs [42]. Activated carbon has also been widely used to effectively remove pollutants in MWSSs [73]. However, activated carbon has been reported to be less effective at removing metals and some inorganic pollutants—especially FCN—due to their low absorbability in poor quality wastewater. It was reported that the adsorption capability of activated carbon depends on the potpourri of available chemical species, thus some research has suggested modification of different activated carbon functional groups to enhance selective adsorption capability [74]. The use of such activated carbon can result in increased production cost, which would in turn increase operational costs of SNaD, thus making this option a less desirable remedial strategy for TN reduction when considering the inhibition of FCN. Thus, more appropriate and less costly methods are required, with some biological processes being proposed as suitable approaches [75].

*5.2. Biological Systems Responsible for Lowering FCN Concentration Prior to SNaD*

As a remedial strategy, the elimination of FCN by microbial processes carried-out during wastewater treatment is usually employed to detoxify FCN into $NH_4$-N. These microorganisms use different mechanisms for FCN degradation with five different FCN degradation mechanisms known, which are hydrolytic, oxidative, reductive, substitution/transfer, and synthesis pathways [65]. The hydrolytic, the oxidative, and the reductive pathways are because of enzymatic actions for which FCN is transformed into simple organic or inorganic byproducts such as $NH_4$-N and $CO_2$, and the other two mechanisms (substitution/transfer and synthesis mechanisms) are responsible for the assimilation of FCN [65].

These pathways are used for the assimilation of FCN as a nitrogen and a carbon source. The hydrolytic pathway is catalyzed by five different enzymes, including cyanide hydratase, nitrile hydratase, and thiocyanate hydrolase. These enzymes have specific activators for and direct hydrolysis of FCN. Additionally, some hydrolyze the triple bond between the carbon and the nitrogen elements to form formaldehyde. Others, including nitrilase and cyanidase, are effective in the microbial metabolic activity and the conversion of FCN into $NH_4$-N and a carboxylic acid [60,72].

The oxidative pathway involves oxygenolytic conversion of the FCN into $CO_2$ and $NH_4$-N; although, this pathway requires an addition of a carbon source, e.g., agricultural waste extracts, and nicotinamide adenine dinucleotide phosphate (NADPH) to catalyze the degradation pathway [60,72]. Moreover, the oxidative pathway is divided further into two distinctive pathways involving three enzymes, namely, cyanide monooxygenase, cyanase, and cyanide dioxygenase. The reductive pathway occurs anaerobically and is catalyzed by nitrogenase to convert FCN to methane and ammonium [76], a process that is not facilitated in SNaD.

The substitution/transfer pathway catalyzes FCN assimilation for growth purposes with the aid of rhodenase and mercaptopyruvate sulfurtransferase by using FCN as a nitrogen source. The synthesis pathway is another FCN assimilation pathway that involves the production of an amino acid, β-cyanoalanine, and γ-cyano-α-aminobutyric acid, using other amino acid residues as precursors that react with the FCN compound [76]. Conversely, FCN degradation has been found to be significantly inhibited by some by-products of $NH_4$-N oxidation, such as those analogous to organic acids [77].

To date, there is still minimal literature on the exploitation of FCN resistant or tolerant organisms with an ability to mediate the inhibition effect of FCN compounds in MWSSs. Additionally, Mekuto et al. [78] also reported SNaD at 100–300 mg FCN/L loading by *Bacillus* species. According to the authors, whilst the use of cyanide degrading bacteria to lower toxicity levels of FCN is environmentally benign, the additional reactors in series prior to SNaD can be beneficial for FCN degradation, which can escalate operational costs in MWSSs.

Some FCN degrading microorganisms displayed the ability to degrade FCN subsequent to SNaD. This led to Kim et al.'s [5] proposition of using FCN degrading microorganisms for SNaD, which is an interesting phenomenon that promotes the simultaneous removal of the FCN compound and TN and eventually results in the implementation of SNaD in lower operational cost associated settings, even in an FCN biodegradation reactor.

### 5.3. Overall Remarks on Remedial Strategies in Place to Mitigate FC in SNaD

Although efforts have been made to address the inhibition of SNaD by FCN, the current strategies in place have their limitations; for example, the activated carbon is not effective in the absorption of FCN. Hence, this option is not an appropriate remedial strategy to lower FCN in wastewater on a large scale. The use of FCN degrading bacteria to lower FCN concentration to acceptable standards prior to SNaD has attracted more attention, since it is an environmentally benign option. However, this option can result in the escalation of operational costs associated with the maintenance of the primary reactor designated for FCN degradation. Hence, it is important that this option be re-evaluated to minimize costs.

## 6. A Proposed Sustainable Solution: Environmental Benignity at the Core of SNaD Development

### Application of FCN Resistant Microorganisms in Simultaneous Nitrification and Aerobic Denitrification (SNaD) Under Cyanogenic Conditions

Research has shown that there are FCN resistant microorganisms that can remain active even in concentrations above 18 mg FCN/L [79]. Kim et at [7] successfully achieved SNaD under high FCN conditions using FCN degrading bacteria in a single reactor process [8]. Microorganisms use different mechanisms to resist the influence of FCN through the enzymatic mechanism for FCN degradation through the degradation of FCN into less toxic compounds (cyanotrophic organisms) via different pathways, as previously mentioned, e.g., hydrolytic pathway, oxidative pathway, reductive pathway, substitution/transfer pathway, and synthesis pathway [65,66]. *Pseudomonas pseudoalcaligenes* CECT5344 was sequenced, and it was revealed that four nitrilase genes were responsible for CN-assimilation and six other C-N hydrolase/nitrilase superfamily genes were found in cyanotrophic strains [65]. Nitrilases have been reported to play a role in the nitrogen metabolism of *Colwellia* sp. Arc7-635 [80].

Generally, heterotrophic bacteria that degrade FCN are typically able to assimilate $NH_4$-N, i.e., a byproduct of FCN biodegradation, as a nitrogenous source. Thus, it has been reported that some of the FCN degrading bacteria are also nitrogen assimilators. The number of nitrifying bacteria has been found in FCN rich environments—an indication of the adaption of nitrifying and denitrifying microorganisms to FCN [81]. Ryu et al. [82] reported simultaneous nitrification and thiocyanate (SCN) degradation, demonstrating that FCN and SCN degrading bacteria can be used to mediate the FCN inhibition effect in SNaD systems. Other genes in *P. pseudoalcaligenes* CECT5344 indicated a presence of polyhydroxyalkanoates (PHA) synthesis, which has a potential to biodegrade numerous toxicants, including aromatic compounds such as phenol [50].

Although some scientists have suggested the use of FCN degrading microorganisms to eliminate FCN inhibition on SNaD [60,83,84], more work still needs to be done in order to understand these processes when the wastewater experiences high concentrations of FCN, including other secondary toxicants such as heavy metals (see Table 2). Furthermore, the description of SNaD using numerical models is underdeveloped.

**Table 2.** Studies that have successfully used cyanide degrading microorganisms for nitrification and aerobic denitrification.

| Microorganism | Description of Process Examined | Reference |
|---|---|---|
| *Bacillus* sp | Free cyanide (FCN) biodegradation subsequent nitrification and aerobic denitrification | [78] |
| CN⁻ degrading consortium | Heterotrophic nitrification—aerobic denitrification potential of cyanide and thiocyanate degrading microbial communities under cyanogenic conditions | [84] |
| *Enterobacter* sp., *Yersinia* sp. And *Serratia* sp | Nitrification and aerobic denitrification under cyanogenic conditions | [85] |
| *Pseudomonas fluorescens* | Elimination of cyanide inhibition through cultivation of cyanide degrading bacteria | [8] |
| *Thiobacillus* and *Micractinium* | Simultaneously remove SCN (thiocyanate) and total nitrogen | [82] |

Therefore, proper models that describe the behavior of these FCN degrading bacteria in SNaD, even when performing nitrification including denitrification under high FCN loading conditions, need to be developed and evaluated. Furthermore, the thermodynamics of SNaD under FCN conditions need to be assessed to theoretically elucidate the feasibility of these processes on an industrial scale. This will provide insight into SNaD facilitated by FCN degrading bacteria, which will enable the proper control of SNaD under high FCN conditions.

## 7. Conclusions

SNaD system development faces many challenges; among these is the inhibition of SNaD consortia by FCN, a predominant challenge in most MWSSs. FCN is a by-product of most industrial processes, such as electroplating and ore processing in the mining industry. FCN enters MWSSs through various pathways, which include run-off from cyanide spills or disposal of FCN containing wastewater from numerous industries. Different psycho-chemical methods have been used to treat FCN prior to the SNaD process; however, these methods produce undesirable by-products and they are expensive. Hence, it is important that a sustainable solution to the FCN inhibition of SNaD be developed. Biological removal of FCN has been thoroughly studied, and it is the most commonly used method due to its cost-effectiveness and sustainability. This method has been used as pre-treatment of FCN prior to the influent entering the SNaD; nevertheless, this procedure increases the cost associated with the operation of the SNaD systems.

The ability of FCN degrading microorganisms to carry-out simultaneous nitrification and SCN degradation has also been recommended for SNaD. This approach not only provides a solution to the inhibition of FCN but also provides a solution to the slow growth rate of common SNaD microorganisms. Therefore, the application of FCN degrading microorganisms could provide a sustainable solution to

the inhibition of SNaD by other toxic pollutants and prevent biomass washout. The utilization of FCN resistant or degrading microorganisms for SNaD has been suggested by other scientists. However, for the use of FCN resistant or degrading microorganisms to minimize the inhibition effect of FCN towards processes of TN removal, mathematical and thermodynamic models are required to better understand SNaD as a sustainable approach to eradicating the inhibition effect of FCN in SNaD systems. There is still limited information about the employment of these suitable microorganisms for SNaD; thus, this paper discusses the application of FCN resistant or degrading microorganisms for SNaD to reduce the effect of FCN inhibition, even under conditions whereby agricultural waste can be used as a supplementary nutrient source and as an immobilization surface for improved efficacy of microbial proliferation for the advancement of SNaD.

**Author Contributions:** Conceptualization, N.M., S.K.O.N.; Formal analysis, S.K.O.N., L.C.R., B.S.C. and E.I.O.; Funding acquisition, S.K.O.N.; Investigation, N.M.; Project administration, S.K.O.N., B.S.C. and E.I.O.; Resources, S.K.O.N.; Supervision, S.K.O.N., B.S.C. and E.I.O.; Writing – original draft, N.M.; Writing – review & editing, S.K.O.N., B.S.C. and E.I.O., L.C.R.

**Funding:** Cape Peninsula University of Technology, the University Research Fund (URF RK16), funded this research.

**Acknowledgments:** The authors would like to acknowledge the support given by Cape Peninsula University of Technology Staff and the University Research Fund (URF RK16).

**Conflicts of Interest:** The authors declare that there is no conflict of interest related to this study.

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
