# Peer review of "Sustainable Approach to Eradicate the Inhibitory Effect of Free-Cyanide on Simultaneous Nitrification and Aerobic Denitrification during Wastewater Treatment"

_sustainability, doi:10.3390/su11216180_

Round 1

Reviewer 1 Report

A substantial part of the paper (sections 2 and 3) covers well-known facts (e.g. the operational phases of a sequencing batch reactor in Figure 3) and does not correspond to the title of the paper. Those sections should be minimized, while sections 4 and 5 should be expanded. As it is a review paper, more literature data should be provided and critically discussed.

Author Response

REVIEWER 1

Input 1:

A substantial part of the paper (sections 2 and 3) covers well-known facts (e.g. the operational phases of a sequencing batch reactor in Figure 3) and does not correspond to the title of the paper. Those sections should be minimized, while sections 4 and 5 should be expanded. As it is a review paper, more literature data should be provided and critically discussed.

Response:  Thank you very much for this important input. Indeed, section 3 was condensed and additional  literature added to section 4 and 5. Please refer to the yellow highlited sections.

Reviewer 2 Report

In this study the authors presented the inhibitory effect of free-cyanide on simultaneous nitrification and aerobic denitrification during wastewater treatment. Authors evaluated the application of FCN resistant or degrading microorganism for SNaD to reduce the effect of FCN inhibition. The investigated topic are interesting and deserves to be studied. This study could be important for a number of researchers working in the area of environmental science and wastewater treatment. However, the work needs some enhancements before it can be published. I recommend minor revision of the manuscript based on the following comments:

There are some minor general drawbacks and comments, listed below:

Authors should do a potent literature review and scrutinize the most relevant and recent published papers in high-quality journal articles. However, the literature includes articles from the 1980s and 1990s. Authors should consider referring to newer articles.

Other drawbacks and comments:

Page 6, line 152: Please correct “Figure 1” to “Figure 2”. Please check the figure caption.

Page 7, line 191: Please check the figure number. I think it should be figure 3.

Page 7, line 207: The dot at the end of the sentence is missing.

Page 9, line 292, Eq. 2: Please check the equation 2.

Page 11, line 368: The dot at the end of the sentence is missing.

Author Response

REVIEWER 2

General comments

In this study the authors presented the inhibitory effect of free-cyanide on simultaneous nitrification and aerobic denitrification during wastewater treatment. Authors evaluated the application of FCN resistant or degrading microorganism for SNaD to reduce the effect of FCN inhibition. The investigated topic are interesting and deserves to be studied. This study could be important for a number of researchers working in the area of environmental science and wastewater treatment. However, the work needs some enhancements before it can be published. I recommend minor revision of the manuscript based on the following comments: There are some minor general drawbacks and comments, listed below:

Response: Thank you very much for this critical analysis of our paper.

Input 1

Authors should do a potent literature review and scrutinize the most relevant and recent published papers in high-quality journal articles. However, the literature includes articles from the 1980s and 1990s. Authors should consider referring to newer articles.

Response: Recent journal has been consulted- See list of references consulted below.

Clough, T.J.; Lanigan, G.J.; de Klein, C.A.; Samad, M.S.; Morales, S.E.; Rex, D.; Bakken, L.R.; Johns, C.; Condron, L.M.; Grant, J.; Richards, K.G. Influence of soil moisture on codenitrification fluxes from a urea-affected pasture soil. Sci. Rep 2017, 7(1), 2185. He, T.; Li, Z.; Sun, Q.; Xu, Y.; Ye, Q. Heterotrophic nitrification and aerobic denitrification by Pseudomonas tolaasii Y-11 without nitrite accumulation during nitrogen conversion. Bioresour. Technol 2016, 200, 493-499. Szabó, E., Hermansson, M., Modin, O., Persson, F. and Wilén, B.M., 2016. Effects of wash-out dynamics on nitrifying bacteria in aerobic granular sludge during start-up at gradually decreased settling time. Water 2016, 8(5), 172. Li, G.; Puyol, D.; Carvajal‐Arroyo, J.M.; Sierra‐Alvarez, R.; Field, J.A. Inhibition of anaerobic ammonium oxidation by heavy metals. J. Chem. Technol. Biotechnol 2015, 90(5), 830-837. Aslan, S.; Sozudogru, O. Individual and combined effects of nickel and copper on nitrification organisms. Ecol. Eng 2017, 99, 126-133. Show, K.Y.; Lee, D.J.; Pan, X. Simultaneous biological removal of nitrogen–sulfur–carbon: recent advances and challenges. Biotechnol. Adv 2013, 31(4), 409-420. Zhang, J.; Zhou, J.; Han, Y.; Zhang, X. Start-up and bacterial communities of single-stage nitrogen removal using anammox and partial nitritation (SNAP) for treatment of high strength ammonia wastewater. Bioresour. Technol 2014, 169, 652-657. Luque-Almagro, V.M.; Moreno-Vivián, C.; Roldán, M.D. Biodegradation of cyanide wastes from mining and jewellery industries. Curr Opin Biotechnol 2016, 38, 9-13. Huang, X.; Urata, K.; Wei, Q.; Yamashita, Y.; Hama, T.; Kawagoshi, Y. Fast start-up of partial nitritation as pre-treatment for anammox in membrane bioreactor. Biochem. Eng. J 2016, 105, 371-378. Daverey, A.; Chen, Y.C.; Dutta, K.; Huang, Y.T.; Lin, J.G. Start-up of simultaneous partial nitrification, anammox and denitrification (SNAD) process in sequencing batch biofilm reactor using novel biomass carriers. Bioresour. Technol 2015, 190, 480-486. Daverey, A.; Chen, Y.C.; Sung, S.; Lin, J.G. Effect of zinc on anammox activity and performance of simultaneous partial nitrification, anammox and denitrification (SNAD) process. Bioresour. Technol 2014, 165, 105-110. Gunatilake, S.K. Methods of removing heavy metals from industrial wastewater. Methods 2015, 1(1), 14. Lin, J.; Meng, Y.; Shi, Y.; Lin, X. Complete Genome Sequences of Colwellia sp. Arc7-635, a Denitrifying Bacterium Isolated from Arctic Seawater. Curr. Microbiol 2019,76, 1-5. Watts, M.P.; Moreau, J.W. New insights into the genetic and metabolic diversity of thiocyanate-degrading microbial consortia. Appl. Microbiol. Biotechnol 2016, 100(3), 1101-1108. Ryu, B.G.; Kim, W.; Nam, K.; Kim, S.; Lee, B.; Park, M.S.; Yang, J.W. A comprehensive study on algal–bacterial communities shift during thiocyanate degradation in a microalga-mediated process. Bioresour. Technol 2015, 191, 496-504.

Input 2

Page 6, line 152: Please correct “Figure 1” to “Figure 2”. Please check the figure caption.

Response 2: The correction has been made from “Figure 1 Diagram representing Simultaneous Nitrification and Aerobic denitrification via nitrite route (A).  The representation of floc in activated sludge with aerobic and anoxic zone (B).” TOFigure 2. Diagram representing; (A) different simultaneous nitrification and aerobic denitrification mechanisms as well as simultaneous nitrification and aerobic denitrification via nitrite route. (B) Representation of floc in activated sludge with aerobic and anoxic zone.

Input 3

Page 7, line 191: Please check the figure number. I think it should be figure 3.

Response 3: The correction has been made- The figure number have been changed into figure 3.

Input 4

Page 7, line 207: The dot at the end of the sentence is missing.

Response 4: The correction has been made and dot placed correctly.

Input 5

Page 9, line 292, Eq. 2: Please check the equation 2.

Response 5:Corrected.

Input 6

Page 11, line 368: The dot at the end of the sentence is missing.

Response 6: Corrected.

Reviewer 3 Report

This paper tries an interesting and appropriate topic. However, there is a lack of clarity on the application of methods, which makes it very difficult to adequately evaluate its scientific quality.

A deep improve at methodology description is needed.

Results are clear and allow comprehending the formulated hypothesis through different assessed aspects. Nevertheless, the discussion is poor/missing.

Conclusions are poor concise and are not in concordance with the obtained results, they just repeat the results without a real analysis of the relevance. Perhaps, a deeper description of novelty could be useful to appreciate the relevance of the work

Author Response

REVIEWER 3

Input 1

This paper tries an interesting and appropriate topic. However, there is a lack of clarity on the application of methods, which makes it very difficult to adequately evaluate its scientific quality.

Response 7: Thank you very much for this critical analysis of our paper. Indeed, more literature on the appropriate method and discussion have been added and the method applications were also clarified (please refer to the yellow highlited sections).

Input 1

A deep improve at method and methodology description is needed.

Response 8: More literature on the appropriate method and discussion have been added- see yellow sections.

Input 1

Results are clear and allow comprehending the formulated hypothesis through different assessed aspects. Nevertheless, the discussion is poor/missing.

Response 9: The discussion part has been improved – Please refer to yellow highlited sections.

Input 1

Conclusions are poor concise and are not in concordance with the obtained results, they just repeat the results without a real analysis of the relevance. Perhaps, a deeper description of novelty could be useful to appreciate the relevance of the work

Response 10: The conclusion has been extended and improved accordingly.